# Testing the Development of a Diet-Based Bisphenol a Score to Facilitate Studies on Child Neurodevelopment: A Pilot Project

**DOI:** 10.3390/ijerph22081174

**Published:** 2025-07-25

**Authors:** Marisa A. Patti, Apollo Kivumbi, Juliette Rando, Ashley Song, Lisa A. Croen, Rebecca J. Schmidt, Heather E. Volk, Kristen Lyall

**Affiliations:** 1AJ Drexel Autism Institute, Drexel University, Philadelphia, PA 19104, USAkld98@drexel.edu (K.L.); 2Department of Mental Health, Johns Hopkins Bloomberg School of Public Health, Johns Hopkins University, Baltimore, MD 21205, USA; 3Division of Research, Kaiser Permanente Northern California, Pleasanton, CA 94588, USA; 4Department of Public Health Sciences, The MIND Institute, University of California Davis, Davis, CA 95817, USA; rjschmidt@ucdavis.edu

**Keywords:** Bisphenol A, endocrine disrupting chemicals, gestational diet, autism-related traits

## Abstract

While gestational Bisphenol A (BPA) exposure has been associated with autism, limited work has focused on dietary sources. Here, we sought to develop a summary metric to capture dietary exposure specifically and test its associations with measured levels, as well as child traits related to autism. Participants (*n* = 116) were from the Early Autism Risk Longitudinal Investigation (EARLI) Study, which recruited pregnant women who previously had a child diagnosed with autism. Maternal concentrations of BPA were quantified in urine, and dietary sources of BPA were ascertained via food frequency questionnaires during gestation. A novel BPA “dietary burden score” was developed based on reported intake of foods known to contribute to BPA exposure (i.e., canned foods) from a Dietary History Questionnaire modified for pregnancy. Child autism-related traits were assessed via the Social Responsiveness Scale (SRS-2). We examined associations between BPA biomarkers, dietary burden scores, and child SRS scores. Dietary burden scores were weakly correlated with urinary BPA concentrations (R = 0.19, *p* = 0.05) but were not associated with child SRS scores. Our work suggests that more detailed dietary assessments may be needed to fully capture diet-based BPA exposures and address diet as a modifiable source of chemical exposure to reduce associated health impacts of BPA.

## 1. Introduction

Diet can be a source of beneficial nutrients but also can be a source of harmful environmental contaminants. For example, compounds such as phenols and other plasticizers can leach into food from procedures related to packaging, processing, and preparation [1,2,3]. Diet is a key source of exposure to phenols, including Bisphenol A (BPA), which contributes to the ubiquitous and pervasive exposure of these compounds in the US population, including pregnant individuals [4,5]. Phenols are considered to be endocrine-disrupting chemicals (EDCs), and gestational exposure to these and other EDCs has been previously associated with increases in childhood autism-related traits [6,7,8,9]. Autism spectrum disorder (ASD; hereafter autism), with a current prevalence of 1 in 31 children in the US, is defined by challenges in social communication and restricted and repetitive behaviors [10]. Such behaviors present along a complex spectrum, and evidence supports a continuous distribution of traits that extend beyond the diagnostic boundary to the general population [11,12].

Unlike other sources of EDCs, such as personal care products, consumers are often unaware of the types or levels of chemical contamination in their food, as compounds with potential contamination are not included on nutrition or ingredient labels. Elevated levels of phenol contamination are found in highly processed foods [13,14], which tend to be consumed more frequently among those in lower socioeconomic positions [15,16,17], suggesting a health disparity. While measuring levels of chemicals can be important in documenting exposures, investigating biomarkers alone can present challenges if biospecimens are not available for all populations of interest, or in relevant time windows suspected to have the strongest health effects and may have limited utility for developing targeted interventions if sources of exposure are not clear to the public. For these reasons, research into dietary sources of these chemicals may be useful to identify additional opportunities for intervention, in addition to exposure reduction strategies at the production level [18].

Using data from a longitudinal cohort study, we developed a novel diet-based BPA burden score to test associations with the measured levels and with child autism-related traits. We hypothesized that gestational biomarkers of BPA would positively correlate with higher consumption of key dietary sources of BPA and that the novel diet-based score would be associated with a greater degree of autism-related traits in children.

## 2. Materials and Methods

### 2.1. Study Participants

Participants were drawn from the Early Autism Risk Longitudinal Investigation (EARLI) study, an autism familial cohort that recruited pregnant mothers who previously had a child diagnosed with autism (study proband) and followed the subsequent child through early development until at least age 3. Autism is genetic, contributing to patterns of recurrence within families; however, genetic influences alone do not explain all of the heterogeneity observed within autism, pointing to possible environmental factors or gene–environment interactions [19,20]. The subsequent children (younger siblings of the study proband) are at an increased likelihood of autism and other neurodevelopmental conditions compared to those drawn from the general population, offering a unique opportunity to prospectively evaluate risk factors. These children are the focus of these analyses. Recruitment and data collection details have been previously published [21]. Briefly, of the 806 eligible pregnant women and their biological children with autism, 264 women were recruited between 2009 and 2012. Of note, BPA, the chemical exposure of interest, was banned in some, though not all, consumer products in 2012 [22], indicating that participants in EARLI were potentially still exposed to products containing BPA during the gestational period. Recruitment took place at four sites in the United States: Pennsylvania (Drexel/Children’s Hospital of Philadelphia); Maryland (Johns Hopkins/Kennedy Krieger Institute); and Northern California (University of California, Davis and Northern California Kaiser Permanente). Each participating site’s Institutional Review Board (IRB) approved the EARLI study. Eligibility criteria indicated that women must have previously had a child with an autism diagnosis confirmed by an EARLI clinician, be ≥18 years old, <29 weeks of gestation, be communicative in English or Spanish, and live within 2 h of a study site. Written informed consent was provided by all women and for their children. Among those enrolled, 176 women delivered live-born, singleton infants. We excluded *n* = 60 participants with missing BPA biomarker, dietary data, and covariate information. Our final analytic sample included *n* = 116 mother/child dyads (Appendix A).

### 2.2. BPA Exposure Assessment

Maternal participants provided up to two urine samples, with first morning void urine samples collected during the 1st, 2nd, and/or 3rd trimesters of pregnancy. Most women provided urine during the 1st trimester (*n* = 113), and a second urine sample in the 2nd trimester (*n* = 65); 48 provided a second sample in the 3rd trimester.

Urine samples were stored at ≤−20 °C and sent to the Centers for Disease Control and Prevention (CDC) on dry ice. Laboratory technicians used a modified method of online solid-phase extraction coupled with isotope dilution–high-performance liquid chromatography with tandem mass spectrometry to quantify urinary BPA metabolite concentrations [23]. Each analytic batch includes reagent blanks and low- and high-concentration quality control (QC) materials, which were evaluated using standard statistical probability rules. The CDC laboratory is licensed by the Clinical Laboratory Improvement Act (CLIA) of 1988. Analytical measurements were conducted following strict quality assurance/quality control (QA/QC) guidelines, CLIA guidelines, and frequent proficiency testing. The limit of detection (LOD) for BPA metabolites was 0.20 ng/mL. Values below the LOD were assigned values of the LOD/√2 [24].

To control for individual variation in urine dilution, we standardized BPA concentrations (ng/mL) by urinary creatinine concentrations (µg/g creatinine). As BPA concentrations were right-skewed, we log_−10_ transformed BPA concentrations to reduce the influence of extreme observations [25]. We then took the mean of all available log_−10_ transformed, creatinine-standardized urinary BPA concentrations from the first, second, and/or third trimester for each participant to have one measure to represent the entire gestational period.

### 2.3. BPA Dietary Burden Score

Dietary information during gestation was collected via food frequency questionnaires (FFQs), specifically a modified version of the National Cancer Institute’s Dietary History Questionnaire (DHQ). Modifications included questions regarding organic and canned food intake, with the goal of capturing information about potential toxicant exposure. Dietary information was collected up to two times during gestation: during the 20th week of gestation, with results pertaining to weeks 1–20 of pregnancy, and the 36th week of gestation, with results pertaining to weeks 21–36 of pregnancy. For participants with repeated measures of dietary information, we took the average of responses to represent the entire gestational period. For participants with dietary information for only one time point, we used those responses to represent the mean. Prior work has previously reported strong correlations of self-reported food intake from the first half of pregnancy (1–20 weeks gestation) with the second half of pregnancy (21–36 weeks gestation) [26].

Building upon prior work to assess BPA contamination through diet [27], we selected questions from the full FFQ related to the top sources of BPA contamination (specific items included can be found in Appendix A). We then categorized exposure status to each food item into no exposure (did not consume the food item in question) and terciles of dietary intake. Consistent with prior work, we then provided a weight for each source of exposure to account for differences in BPA contamination per food item (Appendix A) [27,28]. Briefly, weights were derived based on a literature review of biomonitoring studies to determine the range of BPA contamination that occurred across multiple food items [28]. For example, canned foods are the primary contributor of BPA contamination [29,30], and were assigned higher weights relative to other dietary sources, such as heating food in plastic containers [31,32]. Next, we summed each weighted item to determine an individual’s BPA summary burden score. Note, the goal of developing the BPA burden score was not to directly estimate dietary levels of BPA contamination. Rather, we were more interested in estimating rank order differences in dietary-based BPA contamination amongst the EARLI study sample participants, consistent with prior work estimating relative dietary-based BPA exposure [27,28].

### 2.4. Autism-Related Traits Behavioral Assessment

Child autism-related traits were assessed with the preschool form of the Social Responsiveness Scale (SRS-2), a validated and reliable questionnaire that assesses the presence of autism-related behaviors in both the general population and in clinical settings [11,33,34]. The SRS questionnaire includes 65 Likert-style items that measure behaviors associated with individuals with autism. SRS scores are presented as sex-standardized T-scores (mean: 50, standard deviation (SD: 10), where higher scores are indicative of more autism-related behaviors. Caregivers completed the SRS when children were 3 years of age (T-scores, mean: 3.04; SD: 0.07 years). We used SRS T-scores as the outcome for our primary analysis.

### 2.5. Covariates

We adjusted for sociodemographic and maternal factors based on biologic plausibility and a priori knowledge, informed by a directed acyclic graph (DAG) (Appendix A). Sociodemographic information and reproductive health information, including race, age at delivery, household income, education, and parity, were collected via questionnaires administered by trained research staff. We recognize that race and ethnicity are social constructs and not biological variables [35]. Race defines a group of people connected by common descent or origin, and ethnicity defines a group of people with common national or cultural traditions. In adjusting for race and ethnicity, we attempted to capture how racism, segregation, and other systematic constructs may impact the observed associations between chemical exposure, diet, and autism-related traits. While adjusting for race and ethnicity alone does not capture the full effect of these factors on measured associations, we do not have the data necessary to further deconstruct this concept in our adjusted models. We assessed smoking during pregnancy via cotinine concentrations from urine measured by immunoassays. Cotinine concentrations were used as continuous variables in analyses [36]. In primary models, we adjusted for maternal age, maternal education, pre-pregnancy BMI, maternal race and ethnicity, gestational cotinine concentrations, and annual household income given associations between these covariates with diet, BPA levels, and autism outcomes in children [10,37,38,39,40].

### 2.6. Statistical Analyses

#### 2.6.1. Primary Analyses

First, we characterized distributions of study sample characteristics by calculating univariate statistics, including maternal sociodemographic and reproductive factors. We then assessed measures of reproducibility between repeated BPA concentrations and dietary intakes using intraclass correlation coefficients (ICCs).

Next, we evaluated associations between average maternal urinary BPA concentrations with BPA dietary burden scores by calculating correlations and using adjusted linear regression. We then calculated covariate adjusted associations between BPA concentrations and BPA dietary burden scores with SRS T-scores. Primary models were adjusted for maternal age, maternal educational attainment, annual household income, maternal tobacco smoke exposure, race and ethnicity, and pre-pregnancy BMI.

#### 2.6.2. Sensitivity Analyses

In sensitivity analyses, we additionally adjusted primary models for health-related behaviors, including maternal breastfeeding practices, prenatal vitamin intake, and the American Health Eating Index dietary pattern [41], given the observed associations with autism outcomes in children [42,43,44,45]. As cigarette filters are a source of BPA exposure [46], and few participants in the EARLI sample are active smokers, we also conducted analyses restricting the sample to participants who self-reported to be non-smokers. We ran separate models where each factor was added to the previously defined list of covariates.

## 3. Results

Roughly half of all mothers (47%) were 35 years or older at the time of delivery (Table 1). The majority were non-Hispanic (84%), White (69%), with high educational attainment (70% completed college and/or obtained advanced degrees). Over 40% had annual household incomes ≥$100,000. Few mothers reported active smoking during gestation (3%), and roughly a third of mothers had pre-pregnancy obesity (35%). The analytic sample had slightly more boys (54%) compared to girls (46%). The average SRS T-score was 48 (SD: 11), and was higher among boys (mean: 50; SD: 13) compared to girls (mean: 46; SD: 7.2). SRS T-scores were lower with increasing maternal age, educational attainment, and annual household income. Mean SRS T-scores did not vary by maternal race and ethnicity.

Almost all participants had at least two urinary BPA measures (*n* = 110), with the vast majority of values > LOD (Appendix A). Repeated urinary BPA concentrations had poor reproducibility (ICC: 0.48). Roughly half of the participating mothers completed dietary information assessments at both time points (*n* = 66, 49%). Of those with repeated dietary intake information, reproducibility of individual components of BPA dietary burden scores ranged from poor (frequency of consuming pre-packaged food in plastic, ICC: 0.34) to good (frequency of canned vegetable intake, ICC: 0.77) (Appendix A). SRS T-scores were roughly normally distributed (mean: 48, SD: 11), (Appendix A), with few children scoring above clinically relevant cut-points of T-scores ≥60 (*n* = 14, 12%) or ≥75 (*n* = 5, 4%).

BPA dietary burden scores had the potential range of 0 (no dietary-based exposure) to 9.75, with values within the analytic sample ranging from 0.50 to 8.50 (median: 3.25) and values skewed towards lower levels of dietary-based BPA exposure. Across terciles of BPA dietary burden scores, distributions of average gestational urinary concentrations of BPA were only slightly elevated in the highest tercile (mean 1.48 ng/mL in tercile 3 vs. 1.27 ng/mL in tercile 1), which included those with the highest intake of BPA-contaminated foods (Table 2). Maternal gestational urinary concentrations of BPA were weakly positively correlated with BPA dietary burden scores (R = 0.19, *p* = 0.046). The proportion of variance was low (r^2^ = 0.034), indicating that BPA dietary burden scores did not explain a substantial amount of the variation in maternal gestational urinary concentrations of BPA. Maternal gestational urinary concentrations of BPA and BPA dietary burden scores were not associated with SRS T-scores (Table 3).

Additional adjustments for breastfeeding and American Healthy Eating Index scores did not substantially alter the pattern of results (Appendix A). Given that very few participants in the analytic sample were self-reported smokers (or self-reported tobacco smoke exposure) or did not take prenatal vitamins during gestation, we conducted adjusted analyses restricting the sample to those who were non-smokers or took prenatal vitamins during gestation. In analyses with restricted samples, adjusted results did not differ from primary analyses using unrestricted samples.

## 4. Discussion

We investigated the association between prenatal BPA exposure from maternal diet and child autism-related traits. Gestational intake of canned fruits and vegetables was the primary contributor to our novel BPA dietary burden score. We observed higher gestational urinary concentrations of BPA with the highest tercile of BPA dietary burden scores, demonstrating broad proof of concept; however, despite borderline statistical significance, the correlation was relatively weak (r = 0.19, *p* = 0.05). We did not observe an association between this dietary burden score or BPA levels with child SRS T-scores in this cohort of children with an older sibling with autism.

Maternal diet during gestation is critical for promoting long-term health for the developing fetus. While there are many health-related benefits related to nutritional components of diet (e.g., folic acid, omega-3 fatty acids, and choline), it is also known that diet is a source of chemical exposure, including to BPA and other phenols as well as to other endocrine-disrupting chemicals [1,2,3]. For BPA, as this chemical contamination occurs through packaging, processing, and preparation, most consumers are unaware of this source of “hidden chemicals”, in contrast to other sources of exposure, including personal care products, where chemical compositions are often included on product labels. Thus, focusing on sources of exposure, rather than the chemical levels themselves, may provide actionable points of intervention [18]. Indeed, studies have demonstrated that restricting diets to avoid foods high in BPA contamination is successful in reducing urinary BPA concentrations [47,48].

We hypothesized that gestational urinary concentrations of BPA and BPA dietary burden scores would be associated with a greater degree of autism-related traits in children. Our overall null results could be attributed to several factors. First, EARLI is a specialized cohort that recruited pregnant women who previously had a child diagnosed with autism, and followed the child resulting from that subsequent pregnancy. The null associations observed with SRS T-scores could be related to potentially stronger genetic contributions to autism in this familial setting, which may wash out the effects of environmental exposures [49,50]. Alternatively, reporting of autism-related traits in the SRS may also differ in these families who already have an older child with autism. It is possible that recognition and identification of related behaviors and differences in perceptions of these behaviors within families of autistic children could contribute to differences in reporting of autism-related traits in this sample relative to samples drawn from the general population [51,52]. In addition, while prior works support the stability of autism-related traits as measured by the SRS, lower mean scores have been observed on the preschool-age SRS form relative to the school-age form, which may have dampened effect sizes here [53]. Future work may consider replicating this study design within a sample, including older age at assessment. These analyses were conducted within a specialized cohort of children who have older siblings with autism, thus limiting the generalizability of these findings. While we do not suspect differences in consumption of BPA-containing foods or measured urinary levels in this sample relative to the general population, distributions of child SRS scores and reporting of those traits are likely different. Thus, future work may consider replicating these methods in a sample drawn from the general population in order to address these considerations.

Second, levels of BPA were relatively low among this study population, which may be due to the federal ban in 2012 for its use in canned goods and increased consumer awareness of the potential harms of BPA [22]. Both scientific and public concern for the health effects of BPA exposure began prior to the federal ban, including petitions to ban the use of BPA in canned foods by the Environmental Working Group in 2007 [54]. Even though BPA was still frequently used in consumer products during the pregnancy data collection period of the EARLI study, consumers may have been aware of these BPA concerns and taken steps to reduce their exposure. This is evidenced by national trends in BPA levels indicating that urinary concentrations decreased over time from 2003 to 2014, though disparities remained, particularly in regard to food security [55]. This could have contributed to the overall low intake of BPA-containing foods amongst EARLI study participants. Participants of the EARLI study also have high levels of educational attainment and high annual household incomes relative to the general US population, both of which have been associated with lower levels of BPA [55,56].

Third, BPA exposure may have been under-ascertained due to reliance on diet as the only source of exposure. Some work has reported personal care product use to be the driving contributor of individual phenol biomarker levels [57], though some prior work has attributed 60–90% of urinary BPA concentrations to dietary sources [47,48,58]. Another study also reported that the diet is the largest contributor to BPA biomarker levels, relative to household dust, dental sealings, or air, although this study did not consider personal care product use [59]. BPA is also used in cigarette filters, and thus smoking could be another source of BPA exposure [46]; however, very few participants in the analytic sample were active smokers (3%). Further, results from sensitivity analyses where we restricted the sample to exclude active smokers did not substantially differ from those of the primary analyses. Receipt paper is also a source of BPA exposure, as elevated levels have been observed among those with reported frequent handling, such as cashiers [46,60]. Thus, it is possible that some occupational exposures could also contribute to measured urinary concentrations of BPA, and our inability to account for these other sources of exposure could explain the modest correlation observed between dietary sources and measured levels observed here. In our work, because the primary dietary source of BPA is canned foods [47,48,58], we weighted dietary intake of canned fruit and vegetable intake higher relative to other sources of BPA contamination, consistent with prior work [27]. However, levels of BPA in canned foods are variable, even when ascertained from the same food items and production companies [61]. These inconsistencies may further contribute towards measurement error of dietary-based BPA exposure [28]. Additionally, the dietary recall survey used in this study did not differentiate all foods consumed between canned, frozen, or fresh foods. Thus, we could not consider all sources of canned food intake, as was the case with other diet-based BPA burden scores [27,28]. Thus, our summary score was likely an underestimate of total dietary-based BPA intake.

Fourth, canned fruits and vegetables, even as a source of BPA, also carry beneficial nutrients, which can have impacts on child neurodevelopment. It is possible that null associations between BPA summary burden scores and SRS T-scores could be attributed to interactions with beneficial nutrients in the foods that carry BPA exposure, as speculated in prior work on pesticide residue burden scores (PRBS) with fruit and vegetable consumption and child autism related traits [26]. Overall, little is understood regarding the relative contributions of multiple sources of chemical exposures of BPA, as well as other phenols and plasticizer chemicals, especially as to an individual’s internal body burden, presenting opportunities for future research.

Fifth, while we adjusted for several confounders in both primary and sensitivity analyses, we cannot rule out the impacts of residual confounding, including confounding by unmeasured nutritional or environmental factors.

This study has several strengths, including a longitudinal design allowing for both prospectively reported gestational diet and urinary concentrations of BPA. Investigating both biomarkers and their sources from a public health prospective allows for more targeted messaging compared to biomarkers alone, as sources of chemicals may be more easily modifiable through future dietary based interventions [18].

However, there are several limitations that should also be considered in the context of these results. First, we were limited by a small sample size that had minimal variation in BPA concentrations and dietary sources of BPA. Given the sample size limitations, we were unable to consider the impacts of effect measure modification by child sex. Prior research has observed sex-specific effects, where boys are more likely to be diagnosed with autism [62], and may be more susceptible to the effects of BPA compared to girls [63,64].

Second, we considered BPA as the primary exposure of interest, though there are many other endocrine-disrupting chemicals known to contaminate food sources, including phthalates, which are found in highly processed or fast foods [17]. Future work should consider other endocrine-disrupting chemicals, BPA-replacement chemicals, and their mixtures. The 68 (BPF) and Bisphenol S (BPS) are chemical compounds similar to BPA that are used as alternatives in the production of plastics [65]. These replacement chemicals share similar chemical structures and properties with BPA and are also known to have endocrine-disrupting effects [65], and have been associated with similar health outcomes as BPA, including decreases in child cognition [66] and childhood obesity [67].

Third, we were unable to account for the measurement error and under-ascertainment of BPA levels and how they relate to diet-based exposures. BPA is a non-persistent chemical with a half-life of <1 day [68], though exposure tends to be chronic and episodic. Within our sample we observed poor reproducibility in repeated urinary concentrations of BPA over the course of gestation. Though some of this measurement error was reduced, given that the majority of participants provided two urine samples, future work may consider additional repeated measures of BPA over time to further investigate potential windows of vulnerability within the gestational period. Indeed, one prior study ascertained nine spot urine samples over multiple days (two weekdays and one weekend), and compared associations with dietary sources of BPA contamination [28]. While canned food intake was moderately positively correlated with urinary BPA levels, less than 50% of the variability in measured BPA biomarkers was explained by expected dietary sources, which also included fast and packaged food [28]. However, this pilot study was conducted in a small sample, where recruitment occurred between 2012 and 2013, notably after legislation to ban the use of BPA in consumer products [22]. Additionally, we could not directly attribute urinary concentrations of BPA to dietary intake, as the dietary intake forms were completed in reference to long windows of time (i.e., 1–20 weeks of gestation and 20–36 weeks of gestation). Future work may consider 24 h dietary recalls to be collected prior to urine samples, so that biomarkers may more accurately reflect diet-based exposure. Prior research has found 24 h dietary recalls to be useful tools in accurately predicting urinary biomarkers of BPA [28,69,70,71]. The use of food frequency questionnaires could also introduce recall or social desirability bias. While the National Cancer Institute diet history questionnaire FFQ used here was not specifically validated for canned food consumption (as the primary dietary source of BPA contamination), it has been previously validated for dietary intake [72,73,74,75].

Additionally, while dietary burdens may be considered a proxy measure of BPA exposure, which is measured with error, prior work has shown proxy measures may be less susceptible to unmeasured confounding by personal factors [76].

## 5. Conclusions

Diet can be a source of hidden exposure to BPA and other endocrine-disrupting chemicals, which can disproportionally impact those who frequently consume ultra-processed, packaged, and fast foods. In this sample of children who have an older sibling diagnosed with autism, we investigated the role of maternal dietary sources and urinary concentrations of BPA with childhood autism related behaviors. We did not observe evidence that BPA dietary burden scores or urinary concentrations of BPA were associated with child SRS T-scores. Future work should consider how maternal dietary sources of other endocrine-disrupting chemicals, including replacement chemicals, could impact child neurodevelopment, as focusing on key sources of exposure can provide insights that may be used in interventions.

## Figures and Tables

**Table 1 ijerph-22-01174-t001:** Study sample characteristics and mean child Social Responsiveness Scale (SRS) total T-scores according to covariates.

		SRS T-Score
Variable	N (%)	Mean (SD)
Overall	116 (100)	48 (11)
Maternal Age (years)		
<30	13 (11)	55 (13)
30–<35	48 (41)	48 (12)
≥35	55 (47)	46 (8.1)
Maternal Race		
White	80 (69)	48 (11)
Black	8 (7)	49 (16)
Other	28 (24)	48 (9.6)
Maternal Ethnicity		
Non-Hispanic	97 (84)	48 (12)
Hispanic	19 (16)	48 (6.7)
Maternal Education		
High School or Less	13	51 (10)
Some College	33	53 (14)
Completed College	70	45 (7.7)
Annual Household Income		
<$50 k	23 (20)	51 (16)
$50 k–<$100 k	45 (39)	50 (10)
≥$100 k	48 (41)	45 (7.4)
Maternal Smoking Status ^1^		
Non-Active	112 (97)	47 (10)
Active	4 (3)	67 (17)
Pre-Pregnancy BMI (kg/m^2^)		
Normal/Underweight < 25	44 (38)	45 (8.9)
Overweight ≥ 25–<30	31 (27)	47 (9.7)
Obese ≥ 30	41 (35)	45 (8.9)
Parity		
1	55 (47)	50 (12)
2	44 (38)	44 (7.5)
3+	17 (16)	51 (10)
Child Sex		
Boys	63 (54)	50 (13)
Girls	53 (46)	46 (7.2)
BPA Dietary Burden Score		
T1: Low BPA Dietary Intake	32 (28)	45 (6.6)
T2: Moderate BPA Dietary Intake	43 (37)	50 (15)
T3: High BPA Dietary Intake	41 (35)	48 (8.1)

SRS: Social Responsiveness Scale, BMI: Body Mass Index, BPA: Bisphenol A. ^1^ Maternal smoking during pregnancy was based on maternal urinary cotinine concentrations (a metabolite of nicotine) during pregnancy. The cutoff point of 50 ng/mL was used to differentiate between non-smoking and active smoking.

**Table 2 ijerph-22-01174-t002:** Distributions of average maternal urinary BPA concentrations by terciles of BPA dietary burden scores.

	Concentrations of BPA (ng/mL) Median (1st Quartile, 3rd Quartile)		
	Tercile 1	Tercile 2	Tercile 3
	1.27 (0.86, 1.93)	1.27 (0.94, 2.23)	1.48 (1.05, 2.25)
N	32	43	41

BPA: Bisphenol A, ICE: Index for the Concentration at the Extremes. Terciles for summary BPA burden scores were based on the following cut-off scores: Tercile 1 (low dietary-based BPA exposure) <2.75; Tercile 2 (medium dietary-based BPA exposure) ≥ 2.75–<4.25; Tercile 3 (high diet-based BPA exposure) ≥ 4.25.

**Table 3 ijerph-22-01174-t003:** Unadjusted and adjusted associations between BPA dietary burden scores and average gestational urinary log_−10_ BPA concentrations and child SRS T-scores.

	Correlation R (*p*-Value)	Unadjusted β (95% CI)	Adjusted ^1^ β (95% CI)
Association between BPA dietary burden scores and SRS T-scores			
	0.09 (0.35)	0.01 (−0.01, 0.04)	0.00 (−0.03, 0.03)
Association between urinary BPA concentrations and SRS T-scores			
	0.09 (0.34)	0.00 (0.00, 0.01)	0.00 (−0.01, 0.01)

BPA: Bisphenol A, SRS: Social Responsiveness Scale. ^1^ Adjusted for maternal age (continuous), education (high school or less v. some college v. completed college), pre pregnancy BMI (continuous), and additionally maternal race and ethnicity (non-white v. Hispanic and or Black or other race), gestational cotinine concentration (continuous, proxy for tobacco smoke exposure), and annual household income (<$50 k, $50–<$100 k, >$1000 k).

## Data Availability

The EARLI cohort study participates in data sharing through the National Database for Autism Research (NDAR). Inquiries about data request specific to these analyses should be addressed to the Principal Investigators of the EARLI study (HV, KL) and/or the corresponding author (MAP).

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
