# Peer review of "Testing the Development of a Diet-Based Bisphenol a Score to Facilitate Studies on Child Neurodevelopment: A Pilot Project"

_ijerph, 2025, doi:10.3390/ijerph22081174_

Round 1
Reviewer 1 Report
Comments and Suggestions for Authors
Interesting study. The presence of BPA is just as meaningful if not more, than the concentration, given that the kinetics have not been fully evaluated in this study. It is unclear what the urine concentration means given that the timing of the exposure and amount of exposure are not addressed. This may explain your results in relation to your hypothesis. I would not expect the urine levels in the mother at this point to correlate to the childs SRS score, given the study design.
Author Response
Reviewer 1:
- Interesting study. The presence of BPA is just as meaningful if not more, than the concentration, given that the kinetics have not been fully evaluated in this study. It is unclear what the urine concentration means given that the timing of the exposure and amount of exposure are not addressed. This may explain your results in relation to your hypothesis. I would not expect the urine levels in the mother at this point to correlate to the childs SRS score, given the study design.
Thank you for your review of our manuscript. Consistent with other studies, including our prior work, of gestational exposure to BPA (and other non-persistent endocrine disrupting chemicals broadly), we considered averaged urinary concentrations to represent estimated exposure levels for the gestational period.1,2 Indeed, BPA has a short half-life (<24 hours), and is quickly excreted from the body, contributing to inconsistency in measurements over time, even within individuals.3 That being said, relative exposure to BPA tends to be chronic due to patterns of behavior associated with exposure. In the case of this work here, we assume individuals who reported higher consumption of BPA contaminated foods generally consume these foods at higher levels than those who did not report consumption, evidenced by stability of dietary patterns over time – even within pregnant women. We agree that measurement error is likely, given that the time when the food frequency questionnaire was collected was in reference to a larger window of time (e.g., 1-20 weeks gestation) compared to when urine samples were collected (e.g., ~16 weeks). We address this in depth in the discussion section of the manuscript, and suggest that future work would benefit from 24-hour dietary recalls collected prior to urine samples to directly address the concerns for measurement error noted here. Evaluating the kinetics of BPA and urinary excretion in pregnant women is outside the scope of this work.
- Braun JM, Kalkbrenner AE, Just AC, Yolton K, Calafat AM, Sjödin A, Hauser R, Webster GM, Chen A, Lanphear BP. Gestational exposure to endocrine-disrupting chemicals and reciprocal social, repetitive, and stereotypic behaviors in 4- and 5-year-old children: the HOME study. Environ Health Perspect. 2014 May;122(5):513-20. doi: 10.1289/ehp.1307261. Epub 2014 Mar 12. PMID: 24622245; PMCID: PMC4014765.
- Patti MA, Newschaffer C, Eliot M, Hamra GB, Chen A, Croen LA, Fallin MD, Hertz-Picciotto I, Kalloo G, Khoury JC, Lanphear BP, Lyall K, Yolton K, Braun JM. Gestational Exposure to Phthalates and Social Responsiveness Scores in Children Using Quantile Regression: The EARLI and HOME Studies. Int J Environ Res Public Health. 2021 Jan 30;18(3):1254. doi: 10.3390/ijerph18031254. PMID: 33573264; PMCID: PMC7908417.
- Sasso AF, Pirow R, Andra SS, Church R, Nachman RM, Linke S, Kapraun DF, Schurman SH, Arora M, Thayer KA, Bucher JR, Birnbaum LS. Pharmacokinetics of bisphenol A in humans following dermal administration. Environ Int. 2020 Nov;144:106031. doi: 10.1016/j.envint.2020.106031. Epub 2020 Aug 13. PMID: 32798798; PMCID: PMC9210257.
Reviewer 2 Report
Comments and Suggestions for Authors
This manuscript presents a novel approach to capturing dietary exposure to Bisphenol A (BPA) during pregnancy by developing a dietary burden score and explores its association with biomarker measurements and child autism-related traits within the EARLI cohort. The study is timely, given the growing interest in modifiable environmental risk factors for neurodevelopmental outcomes, and addresses an important gap in exposure assessment methodologies.
The development of a diet-based BPA burden score represents an innovative step in exposure science, particularly in the context of neurodevelopmental research. The focus on dietary sources, rather than solely on biomarker levels, is highly relevant for public health interventions. The methods are described in detail, including the rationale for score construction and the limitations inherent to the study design and population. Reporting of sensitivity analyses further increases the credibility of the findings.
Suggested revisions:
- The dietary BPA burden score is an important innovation. However, its weak correlation with urinary BPA (R=0.19) suggests that it might not have captured all relevant sources of exposure. The authors acknowledge this, but further discussion of potential non-dietary sources and their contribution to the total body burden would be valuable in highlighting the need for a comprehensive exposure assessment in future studies.
- Line 41: 1 in 36 is 2020 data, add the current 2022 data is 1 in 31, consider revising it.
- Line 173: typo “heath”
- The weighting scheme for the dietary score is based on literature, however, more details in the main text about how these weights were derived and validated would help readers assess the robustness of the methodological approach.
- The use of food frequency questionnaires (FFQs) could introduce recall bias and misclassification. Consider discussing this in more depth, perhaps referencing validation studies of FFQs for BPA-related foods.
- The sample size is relatively small (n=116), with limited variability in both exposure and outcome. This limits the ability to detect or assess effect modification (e.g., by child sex). Although the authors acknowledge this limitation, it would be helpful to clarify whether a post-hoc power analysis was conducted or if any follow-up analyses were considered to evaluate whether the sample size was sufficient to detect the associations of interest.
- The cohort is enriched for familial autism risk and is not representative of the general population. The authors should emphasize more in the discussion section regarding the generalizability of the study findings.
- Please consider discussing how parental awareness (having an older child with autism) may influence reporting on the SRS-2.
- The absence of association between BPA exposure and child SRS scores is well explained. However, the discussion could benefit from a more nuanced consideration of potential confounders and alternative explanations, such as residual confounding by unmeasured nutritional or environmental factors, or the role of gene-environment interactions in this high-risk cohort.
Author Response
This manuscript presents a novel approach to capturing dietary exposure to Bisphenol A (BPA) during pregnancy by developing a dietary burden score and explores its association with biomarker measurements and child autism-related traits within the EARLI cohort. The study is timely, given the growing interest in modifiable environmental risk factors for neurodevelopmental outcomes, and addresses an important gap in exposure assessment methodologies.
The development of a diet-based BPA burden score represents an innovative step in exposure science, particularly in the context of neurodevelopmental research. The focus on dietary sources, rather than solely on biomarker levels, is highly relevant for public health interventions. The methods are described in detail, including the rationale for score construction and the limitations inherent to the study design and population. Reporting of sensitivity analyses further increases the credibility of the findings.
Suggested revisions:
- The dietary BPA burden score is an important innovation. However, its weak correlation with urinary BPA (R=0.19) suggests that it might not have captured all relevant sources of exposure. The authors acknowledge this, but further discussion of potential non-dietary sources and their contribution to the total body burden would be valuable in highlighting the need for a comprehensive exposure assessment in future studies.
We have expanded on this section in the discussion by adding additional sources of BPA exposure.
Line 304-312: “Third, BPA exposure may have been under ascertained due to reliance on diet as the only source of exposure…BPA is also used in cigarette filters, and thus smoking could be another source of BPA exposure; however, very few participants in the analytic sample were active smokers (3%). Further, results from sensitivity analyses where we restricted the sample to exclude active smokers did not substantially differ from those from primary analyses. Receipt paper is also a source of BPA exposure, as elevated levels have been observed among those with reported frequent handling, such as cashiers. Thus, it is possible that some occupational exposures could also contribute to measured urinary concentrations of BPA, and our inability to account for these other sources of exposure could explain the modest correlation observed between dietary sources and measured levels observed here.”
- Line 41: 1 in 36 is 2020 data, add the current 2022 data is 1 in 31, consider revising it.
Thank you for bringing this to our attention. We have updated the current prevalence estimates and related reference.
Line 38-40: “Autism spectrum disorder (ASD; hereafter autism), with a current prevalence of 1 in 31 children in the US, is defined by challenges in social communication and restricted and repetitive behaviors”
- Line 173: typo “heath”
Thank you for identifying this typo, we have corrected this in the main text.
Line 187-193: “In sensitivity analyses, we additionally adjusted primary models for health related behaviors including maternal breastfeeding practices, prenatal vitamin intake, and the American Health Eating Index dietary pattern”
- The weighting scheme for the dietary score is based on literature, however, more details in the main text about how these weights were derived and validated would help readers assess the robustness of the methodological approach.
We have expanded on this section of the methods text.
Lines 135-136: “Consistent with prior work, we then provided a weight for each source of exposure to account for difference in BPA contamination per food item (Table s1).27 Briefly, weights were derived based on literature review of biomonitoring studies to determine the range of BPA contamination that occurred across multiple food items.”
- The use of food frequency questionnaires (FFQs) could introduce recall bias and misclassification. Consider discussing this in more depth, perhaps referencing validation studies of FFQs for BPA-related foods.
We agree that FFQs can contribute to measurement error. We have expanded on this as a limitation in the discussion section.
Lines 378-382: “Third, we were unable to account for the measurement error and under ascertainment of BPA levels and how they relate to dietary based exposures… The use of food frequency questionnaires could also introduce recall or social desirability bias. While the National Cancer Institute diet history questionnaire FFQ used here was not specifically validated for canned food consumption (as the primary dietary source of BPA contamination), it has been previously validated for dietary intake.”
- The sample size is relatively small (n=116), with limited variability in both exposure and outcome. This limits the ability to detect or assess effect modification (e.g., by child sex). Although the authors acknowledge this limitation, it would be helpful to clarify whether a post-hoc power analysis was conducted or if any follow-up analyses were considered to evaluate whether the sample size was sufficient to detect the associations of interest.
We agree that the small sample size is a limitation of this work, and have identified this in the discussion section. We did not conduct post-hoc power analyses, as they are not informative for interpretation of results1. This manuscript resulted from pilot funds, which were enough to support the early stages of this work. In fact, the first author of this manuscript (Patti) used the findings from this pilot project as preliminary data for a K99/R00 grant application, where she proposed to expand on the concept of dietary sources of chemical exposure and child autism outcomes in a larger sample with more comprehensive dietary assessment data.
- Heinsberg LW, Weeks DE. Post hoc power is not informative. Genet Epidemiol. 2022 Oct;46(7):390-394. doi: 10.1002/gepi.22464. Epub 2022 Jun 1. PMID: 35642557; PMCID: PMC9452450.
- The cohort is enriched for familial autism risk and is not representative of the general population. The authors should emphasize more in the discussion section regarding the generalizability of the study findings.
We have expanded on this point in the discussion section.
Lines 277-282: “Future work may consider replicating this study design within a sample, including older age at assessment. These analyses were conducted within a specialized cohort of children who have older siblings with autism, thus limiting the generalizability of these findings. While we do not suspect differences in consumption of BPA containing foods or measured urinary levels in this sample relative to the general population, distributions of child SRS scores and reporting of those traits are likely different. Thus, future work may consider replicating these methods in a sample drawn from the general population in order to address these considerations.
- Please consider discussing how parental awareness (having an older child with autism) may influence reporting on the SRS-2.
We agree that parental awareness may impact reporting of autism-related traits in children. We have expanded on this in the discussion.
Lines 269-273: “Alternatively, reporting of autism-related traits in the SRS may also differ in these families who already have an older child with autism. It is possible that recognition and identification of related behaviors and differences in perceptions of these behaviors within families of autistic children could contribute to differences in reporting of autism-related traits in this sample relative to samples drawn from the general population.”
- The absence of association between BPA exposure and child SRS scores is well explained. However, the discussion could benefit from a more nuanced consideration of potential confounders and alternative explanations, such as residual confounding by unmeasured nutritional or environmental factors, or the role of gene-environment interactions in this high-risk cohort.
We have expanded on these points in the discussion section.
Lines 333-335: “Fifth, while we adjusted for several confounders in both primary and sensitivity analyses, we cannot rule out the impacts of residual confounding, including confounding by unmeasured nutritional or environmental factors.”
Reviewer 3 Report
Comments and Suggestions for Authors
Line 33 – For “Diet is a source of beneficial nutrients, but also can be a source of harmful environmental contaminants” I suggest rephrasing it as diet can be a source of beneficial nutrients depending on proportions, namely as regards to sugars or salt.
Despite mentioned in the discussion, a brief mention could be done to bisphenol F and their potential or not influence on autism, a substitute of bisphenol A in countries where severe restrictions on bisphenol A exist. “ Bisphenol F (BPF) is a chemical compound similar to Bisphenol A (BPA) that is used as an alternative in the production of plastics. It shares similar chemical structures and properties with BPA and is known to have endocrine-disrupting effects, posing potential health risks to humans and the environment.”
Despite the recognized influence of mother’s bisphenol A exposure on autism, it is described that autism is hereditary and therefore does run in families. A majority of autism cases can be linked to inherited genetic mutations. The remaining cases likely stem from non-inherited mutations. The sampled population were pregnant woman which had already children with autism. This is mentioned in the discussion but the authors should briefly mention this in the methods section
Lile 107 – where is written “urinary phthalate metabolite concentrations” the authors should explain better this procedure and explain the choice of these obliquus plasticizer in terms of methodology.
Line 138 – “Child autism-related traits were assessed…”, in which children? The first ones from the pregnant women before the study or the children born during the study? The same for line 183
Line 152 – “We recognize that race is a social construct and not a biological variable. In adjusting for race, we are attempting to capture 152 how racism, segregation, and other systematic constructs may impact the observed asso-153 ciations between chemical exposure, diet, and autism-related traits.” Having autism also a genetic influence, I wonder if this is a correct approach. Please consider “Maternal race/ethnicity and nativity are associated with offspring’s AD diagnosis and severity” Becerra TA, von Ehrenstein OS, Heck JE, Olsen J, Arah OA, Jeste SS, Rodriguez M, Ritz B. Autism spectrum disorders and race, ethnicity, and nativity: a population-based study. Pediatrics. 2014 Jul;134(1):e63-71. doi: 10.1542/peds.2013-3928. Please consider this in the text. Even, the authors correctly considerer factor such as smoking and ethnicity as covariates.
Please explain better the difference between race and ethnicity to clarify the reader and explain in the context of this work the choice for considering these distinct concepts. https://www.lawsociety.org.uk/topics/ethnic-minority-lawyers/a-guide-to-race-and-ethnicity-terminology-and-language
Line 171 – Please justify pre-pregnancy BMI as confounder on the light of a potential relationship with autism. Getz KD, Anderka MT, Werler MM, Jick SS. Maternal Pre-pregnancy Body Mass Index and Autism Spectrum Disorder among Offspring: A Population-Based Case-Control Study. Paediatr Perinat Epidemiol. 2016 Sep;30(5):479-87. doi: 10.1111/ppe.12306.
The authors should also explain the factor “taking vitamins” in the context of the work. Levine SZ, et al. Association of maternal use of folic acid and multivitamin supplements in the Periods Before and During Pregnancy with the Risk of Autism Spectrum Disorder in Offspring. JAMA Psychiatry. doi:10.1001/jamapsychiatry.2017.4050.
Line 242 – Please substitute “in reducing urinary concentrations” per ”in reducing BPA urinary concentrations”
Author Response
Reviewer 3
- Line 33 – For “Diet is a source of beneficial nutrients, but also can be a source of harmful environmental contaminants” I suggest rephrasing it as diet can be a source of beneficial nutrients depending on proportions, namely as regards to sugars or salt.
Thank you for this note, we have updated the manuscript text to reflect this.
Line 31: “Diet can be a source of beneficial nutrients, but also can be a source of harmful environmental contaminants.”
- Despite mentioned in the discussion, a brief mention could be done to bisphenol F and their potential or not influence on autism, a substitute of bisphenol A in countries where severe restrictions on bisphenol A exist. “ Bisphenol F (BPF) is a chemical compound similar to Bisphenol A (BPA) that is used as an alternative in the production of plastics. It shares similar chemical structures and properties with BPA and is known to have endocrine-disrupting effects, posing potential health risks to humans and the environment.”
We agree that more attention should be given to BPA replacement chemicals in future work. We have expanded on this point in the discussion section.
Lines 351-355: “Future work should consider other endocrine disrupting chemicals, replacement chemicals of BPA, and their mixtures. For example, Bisphenol F (BPF) and Bisphenol S (BPS) are chemical compounds similar to BPA that are used as alternatives in the production of plastics. These replacement chemicals share similar chemical structures and properties with BPA and are also known to have endocrine-disrupting effects, and have been associated with similar health outcomes as BPA including decreases in child cognition and childhood obesity.”
- Despite the recognized influence of mother’s bisphenol A exposure on autism, it is described that autism is hereditary and therefore does run in families. A majority of autism cases can be linked to inherited genetic mutations. The remaining cases likely stem from non-inherited mutations. The sampled population were pregnant woman which had already children with autism. This is mentioned in the discussion but the authors should briefly mention this in the methods section
We have expanded on this in the methods section when describing the study participants.
Lines 68-75: “Participants were drawn from the Early Autism Risk Longitudinal Investigation (EARLI) study, an autism familial cohort that recruited pregnant mothers who previously had a child diagnosed with autism (study proband), and followed the subsequent child through early development until at least age 3. Autism is genetic in nature, contributing to patterns of recurrence within families, however, genetic influences alone do not explain all of the heterogeneity observed within autism, pointing to possible environmental factors or gene by environment interactions. The subsequent children (younger siblings of the study proband) are at an increased likelihood for autism and other neurodevelopmental conditions compared to those drawn from the general population, offering a unique opportunity to prospectively evaluate risk factors.”
- Lile 107 – where is written “urinary phthalate metabolite concentrations” the authors should explain better this procedure and explain the choice of these obliquus plasticizer in terms of methodology.
Thank you for bringing this line to our attention, and identifying a typo in the methods. We did not consider phthalates in these analyses and have edited the text accordingly.
Line 113: “We then took the mean of all available log-10 transformed, creatinine-standardized urinary BPA concentrations from the first, second, and/or third trimester for each participant to have one measure to represent the entire gestational period”
- Line 138 – “Child autism-related traits were assessed…”, in which children? The first ones from the pregnant women before the study or the children born during the study? The same for line 183
The study children born to the pregnant women were the focus of these analyses. This has been clarified in the methods.
Lines 75-76: “The subsequent children (younger siblings of the study proband) are at an increased likelihood for autism and other neurodevelopmental conditions compared to those drawn from the general population, offering a unique opportunity to prospectively evaluate risk factors. These children are the focus of these analyses.”
- Line 152 – “We recognize that race is a social construct and not a biological variable. In adjusting for race, we are attempting to capture 152 how racism, segregation, and other systematic constructs may impact the observed asso-153 ciations between chemical exposure, diet, and autism-related traits.” Having autism also a genetic influence, I wonder if this is a correct approach. Please consider “Maternal race/ethnicity and nativity are associated with offspring’s AD diagnosis and severity” Becerra TA, von Ehrenstein OS, Heck JE, Olsen J, Arah OA, Jeste SS, Rodriguez M, Ritz B. Autism spectrum disorders and race, ethnicity, and nativity: a population-based study. Pediatrics. 2014 Jul;134(1):e63-71. doi: 10.1542/peds.2013-3928. Please consider this in the text. Even, the authors correctly considerer factor such as smoking and ethnicity as covariates.
We feel it important to make the distinction that race is a social construct and not a biologic variable, consistent with the most recent recommendations by the American Medical Association. Indeed, differences in measured levels of chemical exposure, dietary intake patterns, and prevalence of autism have been observed across racial groups. Consistent with environmental justice research, differences in health outcomes observed across racial groups can likely be attributed to disproportional distributions of exposures (environmental and social), not differences in genetic factors. Historically, autism prevalence has been higher among White children compared to Black children. This has been attributed to increased access to autism diagnostic services and related interventions among White samples with higher socioeconomic position. As barriers to access have decreased, a new trend has emerged, where autism prevalence is now higher among historically marginalized populations. While prior work has demonstrated racial differences in gestational BPA exposure and intake of BPA contaminated foods, including processed foods, we are underpowered to evaluate how race may mediate the associations evaluated here.
- Please explain better the difference between race and ethnicity to clarify the reader and explain in the context of this work the choice for considering these distinct concepts. https://www.lawsociety.org.uk/topics/ethnic-minority-lawyers/a-guide-to-race-and-ethnicity-terminology-and-language
We have expanded on this in the covariates section of the methods text.
Lines 160 -166: “Sociodemographic information and reproductive health information, including race, age at delivery, household income, education, and parity, was collected via questionnaires administered by trained research staff. We recognize that race and ethnicity are social constructs and not biological variables.42 Race defines a group of people connected by common descent or origin, and ethnicity defines a group of people with common national or cultural traditions. In adjusting for race and ethnicity, we are attempting to capture how racism, segregation, and other systematic constructs may impact the observed associations between chemical exposure, diet, and autism-related traits. While adjusting for race and ethnicity alone does not capture the full effect of these factors on measured associations, we do not have the data necessary to further deconstruct this concept in our adjusted models.”
- Line 171 – Please justify pre-pregnancy BMI as confounder on the light of a potential relationship with autism. Getz KD, Anderka MT, Werler MM, Jick SS. Maternal Pre-pregnancy Body Mass Index and Autism Spectrum Disorder among Offspring: A Population-Based Case-Control Study. Paediatr Perinat Epidemiol. 2016 Sep;30(5):479-87. doi: 10.1111/ppe.12306.
We have added citations to justify pre-pregnancy BMI, among other covariates, that we considered as confounders in adjusted models.
Lines: 170-173: “We adjusted for sociodemographic and maternal factors based on biologic plausibility and a priori knowledge, informed by a directed acyclic graph (DAG) (Supplemental Figure 2)… In primary models, we adjusted for maternal age, maternal education, pre-pregnancy BMI, maternal race and ethnicity, gestational cotinine concentrations, and annual household income given associations between these covariates with diet, BPA levels, and autism outcomes in children.”
- The authors should also explain the factor “taking vitamins” in the context of the work. Levine SZ, et al. Association of maternal use of folic acid and multivitamin supplements in the Periods Before and During Pregnancy with the Risk of Autism Spectrum Disorder in Offspring. JAMA Psychiatry. doi:10.1001/jamapsychiatry.2017.4050.
We have expanded on this section of the methods and provided citations to justify our interest in these additional covariates.
Lines 189-192: “In sensitivity analyses, we additionally adjusted primary models for health related behaviors including maternal breastfeeding practices, prenatal vitamin intake, and the American Health Eating Index dietary pattern,44 given observed associations with autism outcomes in children. As cigarette filters are a source of BPA exposure, and few participants in the EARLI sample are active smokers, we also conducted analyses restricting the sample to participant who self-reported to be non-smokers”
- Line 242 – Please substitute “in reducing urinary concentrations” per ”in reducing BPA urinary concentrations”
This has been clarified in the manuscript text.
Lines 246-248: “Indeed, studies have demonstrated that restricting diets to avoid foods high in BPA contamination is successful in reducing urinary BPA concentrations.”